

# *Miconia lucenae* (Melastomataceae), a new species from montane Atlantic Forest in Espírito Santo, Brazil

Renato Goldenberg[1], Marcelo Reginato[2] and Fabián A. Michelangeli[3]

[1] Departamento de Botânica, Universidade Federal do Paraná, Curitiba, Paraná, Brazil
[2] Departamento de Botânica, Universidade Federal do Rio Grande do Sul, Porto Alegre, Rio Grande do Sul, Brazil
[3] Institute of Systematic Botany, The New York Botanical Garden, New York, United States of America

## ABSTRACT

We describe *Miconia lucenae* R.Goldenb. & Michelang., a new species from the montane Atlantic Forest in Santa Teresa in the state of Espírito Santo. Our analysis, based on three plastid spacers (*atpF-atpH*, *psbK-psbl* and *trnS-trnG*), one plastid gene (*ndhF*, not available for *M. lucenae*), and two nuclear ribosomal loci (nrITS and nrETS), showed that it belongs to a small clade with *Miconia paradoxa* (Mart. ex DC.) Triana (Minas Gerais) and *M. michelangeliana* R.Goldenb. & L.Kollmann (Espírito Santo). The three species in the "Paradoxa clade" can be recognized by the plants with glabrous or glabrescent branches and leaves, white petals and yellow stamens, these with the connectives not prolonged below the thecae, ventrally unappendaged, dorsally unappendaged or with a minute tooth, the latter bilobed or not, glabrous ovary, and the fruits with a persistent calyx. *Miconia lucenae* can be recognized, among the species in this clade, by the shrubby plants with terete young branches, short inflorescences, usually with red axes, and the 2-bracteolate, sessile, 4-merous flowers, with a ciliolate inner portion of the sepals, lanceolate petals, and 4-celled ovaries. This species can be considered as endangered according to IUCN criteria.

# INTRODUCTION

*Miconia* Ruiz & Pav. has nowadays about ca. 1900 species native to the Neotropics (*Michelangeli et al., 2016*). Its circumscription has been recently modified (see *Michelangeli et al., 2016*; *Michelangeli et al., 2019*), and now it is equivalent to the whole tribe Miconieae, in its modern definition (*Michelangeli et al., 2004*; *Michelangeli et al., 2008*; *Goldenberg et al., 2008*; see also *Penneys et al., 2010*; *Michelangeli et al., 2011*). This new circumscription includes former *Miconia* sensu stricto and several other genera, such as *Leandra* Raddi, *Clidemia* D.Don, *Ossaea* DC. and *Tococa* Aubl. Some of these genera or parts of these genera may be monophyletic (*Reginato & Michelangeli, 2016*), but their recognition renders *Miconia* s.s. paraphyletic. For an alternative opinion on this broad circumscription of *Miconia* see *Kriebel (2016)* and *Reginato (2016)*.

Corresponding author
Renato Goldenberg, rgolden@ufpr.br

In the course of floristic work in the state of Espírito Santo, Brazil, we collected an undescribed species with lanceolate petals and terminal inflorescences. In *Cogniaux*'s (*1891*) classification, this species would have been placed in *Leandra*. However, the general floral and vegetative morphology of this species makes any comparison to other species previously placed in *Leandra* in the Atlantic Forest very difficult. Moreover, preliminary data placed this species in traditional *Miconia* s.s.

Even though nowadays this species would be unequivocally placed in *Miconia* s.l., following its modern circumscription, we have opted to present here the description of the new species along with a simplified phylogeny based on molecular markers, in order to explain its phylogenetic placement and better understand its unique combination of morphological characters.

## MATERIALS & METHODS

**Taxonomy.** Specimens from the new species and related ones have been studied in loco in and in the herbaria MBML, NY, RB, UPCB. The specimens from VIES have been checked through images available in virtual herbaria (http://reflora.jbrj.gov.br). All morphological analyses were based on herbarium specimens; floral parts were rehydrated from dried specimens.

**Phylogeny.** Taxon sampling was based on previous phylogenies that sampled the tribe Miconieae (*Goldenberg et al., 2008*; *Martin et al., 2008*). For each previously recognized major clade up to six species were selected and their sequences downloaded from GenBank. We did not keep the traditional generic classification for Miconieae, based on *Cogniaux (1891*; see also *Michelangeli et al., 2004*; *Michelangeli et al., 2008*; *Goldenberg et al., 2008*; *Reginato & Michelangeli, 2016*); i.e., we showed all the names transferred to a single genus, *Miconia* s.l., as proposed in *Michelangeli et al.* (*2016*; *Michelangeli et al. 2019*). The old names are listed in Table S1.

Sanger based DNA sequences of *M. lucenae* (voucher Goldenberg 889) were generated for five molecular markers included in those studies. Total genomic DNA was isolated from silica-dried or herbarium material using the DNeasy Plant Mini Kit (Qiagen, Valencia, CA, USA) following the protocol suggested by *Alexander et al. (2007)*. The molecular data set included three plastid spacers (*atpF-atpH*, *psbK-psbI* and *trnS-trnG*), and two nuclear ribosomal loci (the internal and external transcribed spacers nrITS and nrETS). The amplification protocols and primers used are described in *Reginato & Michelangeli (2016)*. Cycle sequencing was performed with the same forward and reverse primers used for amplification at the high-throughput sequencing service at the University of Washington (USA). Contigs were assembled with Sequencher 4.9 (GeneCodes Corp., Ann Arbor, MI, USA). An additional plastid gene (*ndhF*) available for most of the sampled taxa was also included in the phylogenetic analysis. Genbank accessions of all taxa analyzed are available in the supplementary Table S1.

Sequence alignment was performed with MAFFT v.7 using the strategy G-INS-i (*Katoh, 2013*). DNA substitution models for each of the six makers were selected using jModeltest v.2.1 (*Darriba et al., 2012*), using the 3 model scheme with or without four discrete

rate categories approximating a gamma distribution (+G) and including models with equal/unequal base frequencies (+F). The likelihoods were calculated using a Maximum Likelihood optimized base tree with NNI topology search using phyml (*Guindon & Gascuel, 2003*) and the models were evaluated using the BIC criterion. Tree inference was performed using a Bayesian framework implemented in the program BEAST v.2.5.0 (*Bouckaert et al., 2014*). The analysis was performed using the DNA models recovered in the previous step: GTR (atpF-atpH, psbK-psbI); GTR+G (nrETS, nrITS, trnS-trnG); and HKY+G (*ndhF*). Clock and tree models were linked across markers, the molecular clock prior was set to the lognormal uncorrelated, and the tree prior was set to the Birth and Death model, without calibration points. Two independent runs of 50 million generations each, sampling every 1,000 generations with a 10% burn in were performed. Convergence was assessed using Tracer v.1.5 (*Rambaut & Drummond, 2007*), and runs had ESS values greater than 200 for all parameters. The stable posterior distributions of the independent runs were combined using LogCombiner v.2.5.0 and a maximum clade credibility tree summarized with TreeAnnotator v.2.5.0 (*Bouckaert et al., 2014*).

**Niche modeling.** The potential distribution of *M. lucenae* under current climatic conditions was modeled and evaluated in Maxent 3.4.0 (*Phillips & Dudík, 2008*) using the R package dismo (*Hijmans et al., 2017*). The climatic model was based on its known localities and the 19 climatic variables from the WorldClim data set v.2 with 30" spatial resolution (*Fick & Hijmans, 2017*). The area under the curve (AUC) of the receiver operating characteristic (ROC) was used as evaluation criterion, and all parameters were left as the default.

**Conservation status.** Our assessments were based on range size (criterion B), according to the guidelines of the IUCN (*IUCN, 2012*; *IUCN Standards and Petitions Subcommittee, 2017*). Area of Occupancy (AOO) and Extent of Occurrence (EOO) where calculated using the GeoCat tool (*Bachman et al., 2011*) using the same localities used for Niche modeling.

**SEM.** Seeds and leaves for the SEM images were obtained from herbarium specimens and manually cleaned. The structures were mounted on aluminum stubs, coated with gold-palladium for 2 min in a Hummer 6.2 (Aratech LTD), and examined using a JEOL –JSM 5410LV SEM, with the software JEOL ORION 5410, version 1.72.01 (1999–2004).

The electronic version of this article in Portable Document Format (PDF) will represent a published work according to the International Code of Nomenclature for algae, fungi, and plants (ICN), and hence the new names contained in the electronic version are effectively published under that Code from the electronic edition alone. In addition, new names contained in this work which have been issued with identifiers by IPNI will eventually be made available to the Global Names Index. The IPNI LSIDs can be resolved and the associated information viewed through any standard web browser by appending the LSID contained in this publication to the prefix "http://ipni.org/". The online version of this work is archived and available from the following digital repositories: PeerJ, PubMed Central, and CLOCKSS.

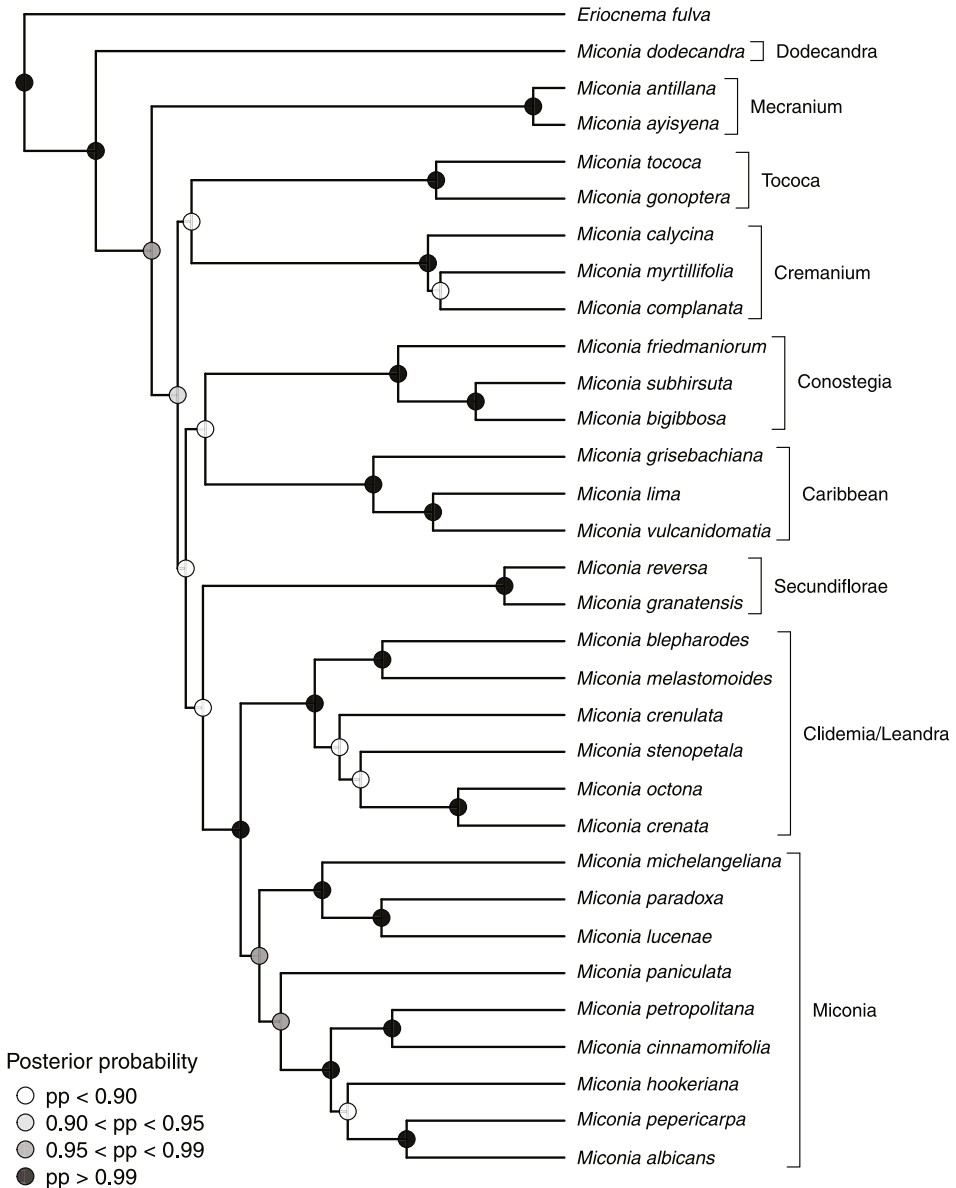

**Figure 1** **Maximum clade credibility tree from the stable posterior distribution (BEAST), including the newly described *M. lucenae* and representatives of major clades in tribe Miconieae.** Posterior probabilities values for nodes follow the legend.

## RESULTS

**Phylogenetic relationships.** *Miconia lucenae* was recovered nested in an early divergent subclade of a clade containing Miconia IV and Miconia V (sensu *Goldenberg et al., 2008*), all of them sister to the Clidemia/Leandra clade (Fig. 1). *Miconia lucenae* is resolved in a clade with *M. michelangeliana* R.Goldenb. & L.Kollmann and *M. paradoxa* (Mart. ex DC.) Triana, called "Paradoxa clade" from now on. Among the other two species in Paradoxa

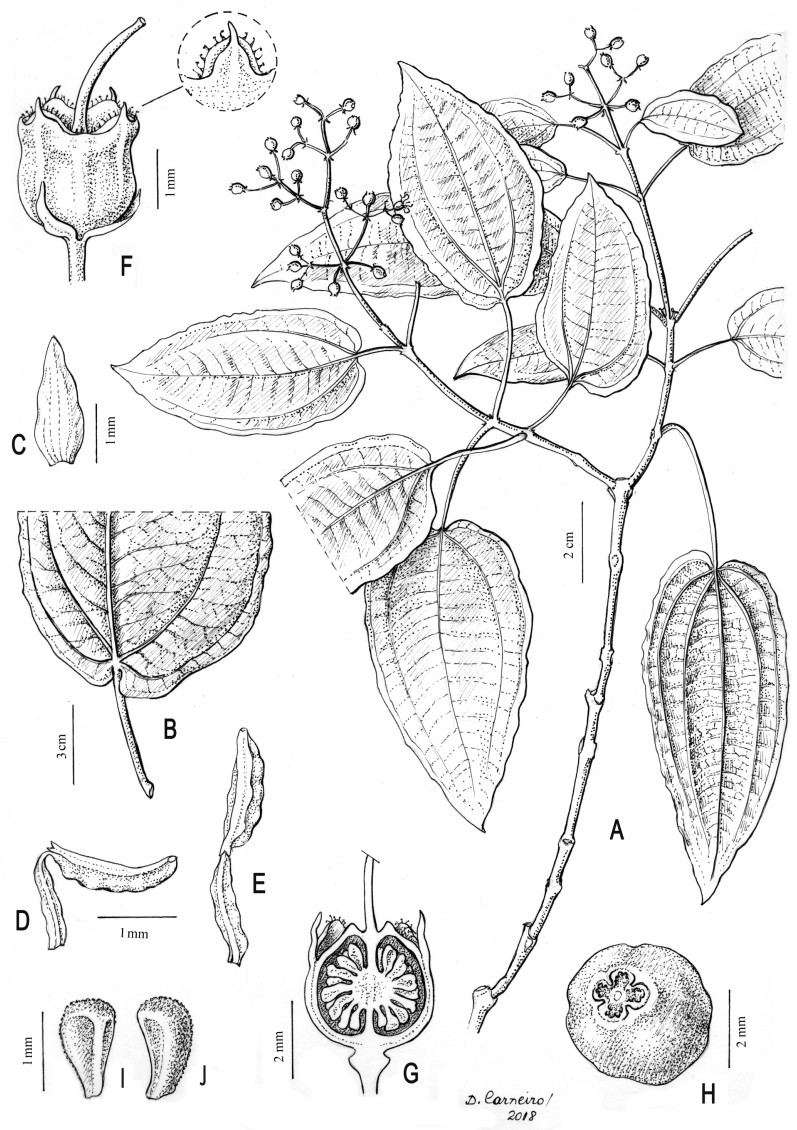

**Figure 2** **Illustration of *Miconia lucenae*.** (A) Fertile branch. (B) Leaf base, abaxial view. (C) Petal, adaxial view. (D) Stamen, lateral view. (E) Stamen, dorsal view. (F) Old flower (petals and stamens removed) with bracteoles, and detail of the sepal, abaxial view. (G) Old flower (petals and stamens removed), longitudinal section. (H) Fruit. (I) and (J) Seeds. A–J from *Goldenberg 1525* (UPCB). Illustration by Diana Carneiro.

clade, *M. lucenae* seems closer to *M. paradoxa* (Fig. 1), which was then chosen as the species to be compared in the formal taxonomic diagnosis.

## Taxonomy

*Miconia lucenae* R.Goldenb. & Michelang., spec. nov.

(Figs. 2, 3 and 4)

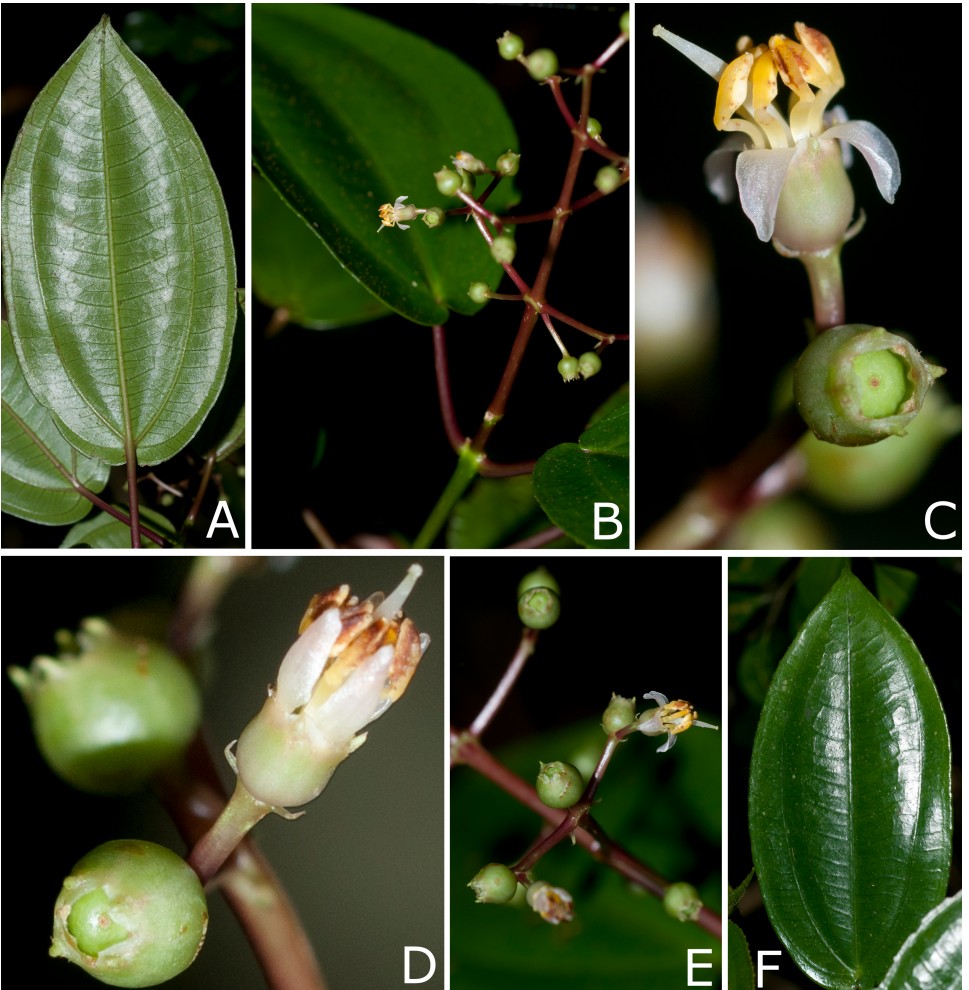

**Figure 3** **Photos of living plants of *Miconia lucenae*.** (A) Leaf, abaxial view. (B) Inflorescence. (C) Flower, lateral view, and young fruit, top view. (D) Old flower (with persistent petals and stamens), lateral view. (E) Inflorescence branch with flowers and young fruits. (F) Leaf, abaxial view. (A–F) from *Goldenberg 1525* (UPCB). Photos by F. Michelangeli.

**Type**: Brazil, Espírito Santo: Santa Teresa, Nova Lombardia, Terreno do Furlani, 19°47′59″S, 40°32′13″W. 900 m. 7 Feb 2011 (fl, fr), *R. Goldenberg. Michelangeli, M.K. Caddah, M. Reginato & L. Kollmann 1525* (holotype: UPCB -71800; isotypes: MBML, NY-02104713, 02104708, RB - 014190053).

**Diagnosis**: *Miconia lucenae* differs from *Miconia paradoxa* in having terete young branches (vs. strongly decussate-flattened in *M. paradoxa*), ciliate inner portion of the sepals (vs. eciliate), and lanceolate petals (vs. obovate).

**Description**: Shrubs 0.5–1.5 m tall. All vegetative parts (including both surfaces of the leaf blades), inflorescences and hypanthia very sparsely and caducously covered with (1) stellate trichomes 0.1–0.3 mm diam, and (2) minute sessile glands, ca. 50 μm long. Young stems terete, slightly swollen at the nodes, these usually with a faint interpetiolar line,

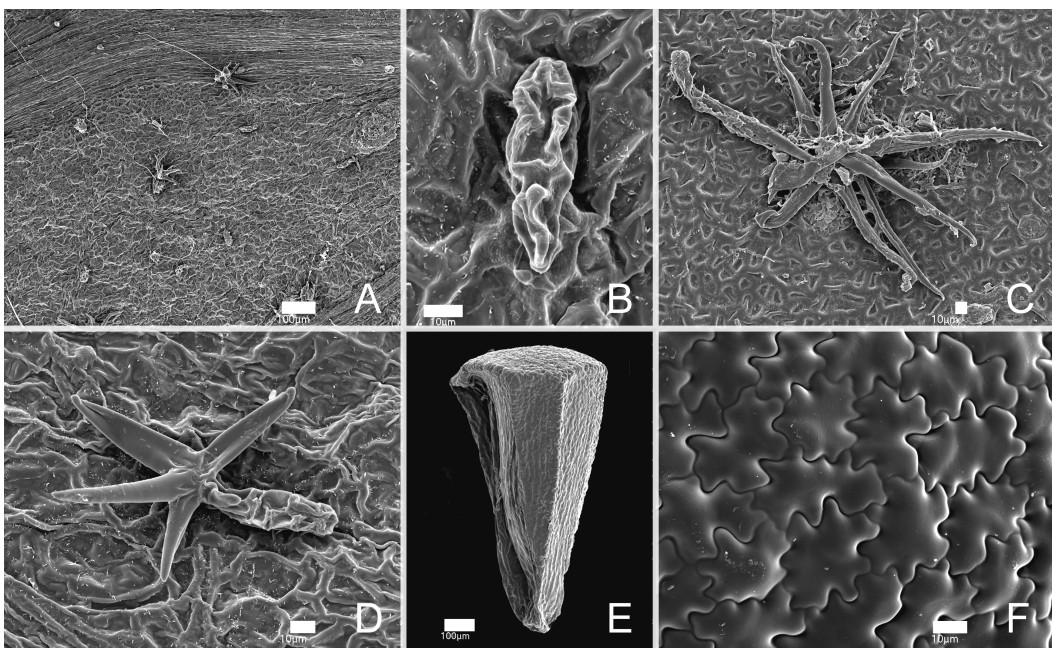

**Figure 4** SEM images of trichomes and seeds of *Miconia lucenae*. (A) Leaf, adaxial surface, with trichomes and sessile glands. (B) Sessile gland. (C–D) Stellate trichomes. (E) Seed, lateral view. (F) Seed, testa surface. All from *L. Kollmann 5594* (NY).

covered with some stellate trichomes when young, then glabrescent. Leaves isophyllous to slightly anisophyllous; petiole 1.5–4.5 cm long, glabrous, reddish; blade 4.5–12 × 2.5–6 cm, oval, elliptic, oval-lanceolate to lanceolate, apex acuminate (seldom acute), base cordate, truncate or obtuse, sometimes strongly oblique, margins undulate or repand, entire, slightly hyaline (seen from below), and eciliate, membranaceous, flat in fresh material but slightly revolute in dried specimens, green in both surfaces (a bit darker on the adaxial surface) in fresh material, markedly discolor in dried specimens, with the adaxial much darker than the abaxial surface; lateral veins strongly to seldom weakly suprabasal (all specimens have leaves with distinct suprabasal nerves, only Goldenberg 1,525 has a few leaves with shortly suprabasal nerves), the inner pair up to seven mm distant from the base, with 2 pairs or seldom 4 pairs, plus and additional, faint, marginal pair, sometimes not symmetrically paired (in leaves with oblique bases), main, lateral and transversal veins plane/impressed, reticulation barely perceptible on the adaxial surface, main, lateral and transversal plane or seldom prominent, reticulation plane but perfectly distinct on the abaxial surface. Panicles 3–6.5 × 2.5–4 cm, terminal, erect, with accessory branches, 2–3 pairs of paraclades, glabrous, the axes reddish; bracts 1–1.5 mm long, linear-subulate, eciliate, caducous; bracteoles 0.8–1.2 × 0.2–0.3 mm, linear-lanceolate, curved upwards, persistent. Flowers sessile, 4-merous. Hypanthium 1.4–2 × 1.8–2 mm, greenish-white at anthesis, greener in older flowers and young fruits, narrowly campanulate to shortly terete, both surfaces glabrous; torus glabrous. Calyx persistent, the tube 0.1–0.2 mm long, greenish-white; sepals with the inner, laminar portion 0.4–0.6 mm

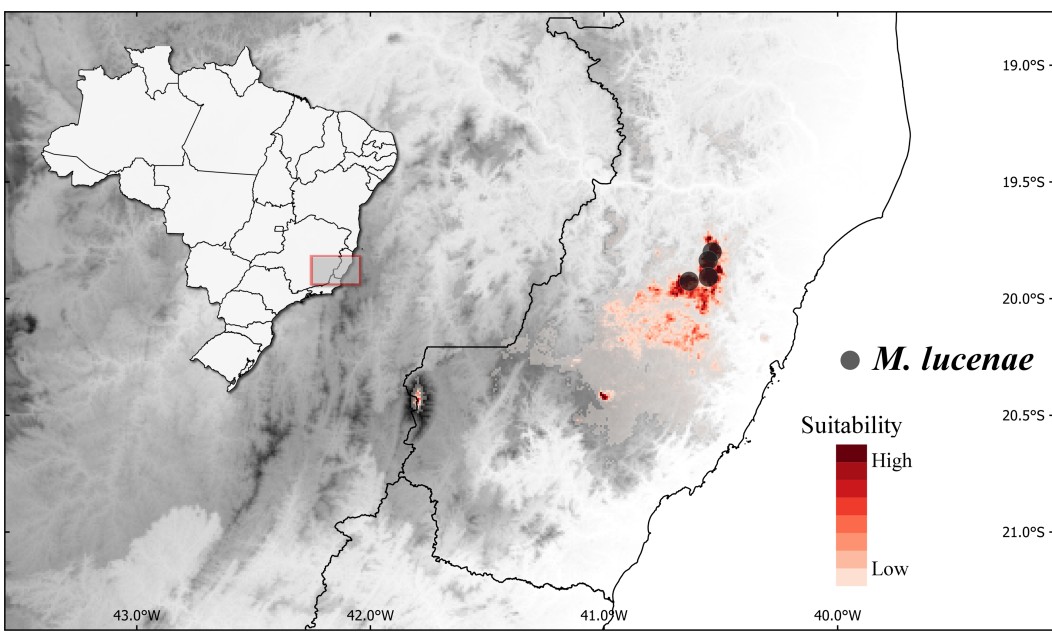

**Figure 5** **Geographic distribution and predicted suitable areas under current climatic conditions of** *Miconia lucenae.* Known distribution is indicated by the point localities and predicted suitable areas under current climatic conditions are in red tones following the legend.

long, greenish-white, erect, hemi-circular, apex rounded, margins sparsely ciliate (the cilia thick, less than 0.1 mm long), otherwise glabrous; outer teeth projecting ca. 0.2 mm above the laminae, light-green, thick-subulate, the apex acute and very shortly mucronulate, glabrous. Petals, 4, 2–2.2 × 0.7–0.8 mm, white, reflexed at anthesis, and apparently erect in old flowers, lanceolate, apex broadly acute to narrowly rounded, margins undulate, glabrous. Stamens isomorphic, erect, arranged in an actinomorphic bundle; filaments 1.3–1.5 mm long, light yellow, glabrous; anthers ca. 1.5–1.7 mm long, light yellow, oblong in ventral view, with the apex slightly archaed backwards, attenuate, with a minute apical to slightly dorsally inclined pore; connective 0.1–0.2 mm prolonged below the thecae, yellow (brighter than filaments and anthers), unappendaged or with two dorsal-basal, minute, less than 0.1 mm long, acute teeth. Ovary ca. 3 × three mm, 4-locular, ca. 2/3 inferior, the free portion projecting ca. one mm, broadly rounded, glabrous; style 2.2–2.7 mm long, filiform, slighly curved, glabrous, stigma punctiform. Berries 3–3.5 × 3–3.5 mm, blackish when ripe, topped with the persistent calyx, glabrous. Seeds 1.1–1.4 × 0.6–0.8 mm, narrowly pyramidate or narrowly oblong, the raphal region narrow and long, (almost 100% of the total seed length), testa rough, minutely tuberculate.

**Distribution and niche modelling.** *Miconia lucenae* has been collected 6 times in 4 different places, all of them in the Municipality of Santa Teresa, state of Espírito Santo (Fig. 5). Only one of the samples has an indication on elevation (900 m), but all of them seem to be collected in places with similar elevation. All specimens were collected in shaded areas, in rainforest understory.

The climatic-based distribution model of *M. lucenae* presented a high value of AUC (0.99). Suitable areas were identified throughout the mountains of Santa Teresa region, especially southern to where the species is known to occur. Additional areas with high suitability includes the "Caparaó" region (41°47′10″W, 20°26′06″ W) and the region of the Municipality of Domingos Martins (41°00′04″W, 20°25′12″S; Fig. 5). Despite the limitations of this model due to the low number of known points for this species, the results can still be informative in the case of collections of this species in new areas.

**Conservation status.** *Miconia lucenae* has an AOO of 50 km$^2$ and an EOO of 16 km$^2$. Given that the six known collections represent four different sites, but all closely located, and the fact that only one of these is inside a conservation area, we recommend that it is considered as "Endangered" following criteria B1B2abiv (*IUCN, 2012*; *IUCN Standards and Petitions Subcommittee, 2017*).

**Paratypes.** Brazil, Espírito Santo: Santa Teresa, São Lourenço, Country Club, 22 Feb 1999, *L. Kollmann, E. Bausen & W. Pizziolo 1973* (MBML); Santa Teresa, Nova Lombardia, Reserva Biológica, Estrada de Goipaba-Açu, 5 Feb 2002 (fr*), L. Kollmann et al. 5484* (MBML, RB, UPCB); Santa Teresa, Nova Lombardia, Reserva Biológica, Estrada para N. Lombardia, 20 Feb 2002 (fr), *L. Kollmann 5594* (MBML, RB, UPCB, VIES); Santa Teresa, Santo Henrique, 22 Jul 2005 (fr), *L. Kollmann & A.P. Fontana 8160* (MBML, UPCB); Santa Teresa, Nova Lombardia, Terreno do Furlani, 13 Jul 2007 (sterile), *R. Goldenberg et al. 889* (MBML, NY, UPCB).

**Etymology.** *Miconia lucenae* honors Dr. Sérgio Lucena Mendes, a primatologist at the Universidade Federal do Espírito Santo, and more than once director of the Museu de Biologia Mello Leitão / Instituto Nacional da Mata Atlântica, in Santa Teresa. This tribute is deserved by his efforts on biological research, conservation policies in the state of Espírito Santo, and, more importantly, on his belief that the general public, and mostly the "capixabas" (i.e., people and things from Espírito Santo) must understand and value one of the most unknown and undervalued treasures that they have in their own backyards: the wondrously diverse Mata Atlântica.

## DISCUSSION

Our phylogenetic analysis recovered the same major clades indentified in previous studies (*Goldenberg et al., 2008*, Michelangeli et al., in prep.). The Paradoxa clade, with *Miconia lucenae*, *M. michelangeliana* and *M. paradoxa*, has not been recognized before. While there is little overall morphological resemblance between *M. lucenae* and the two other species in the Paradoxa clade, all three share glabrous or glabrescent branches and leaves, white petals and yellow stamens, these with connectives not prolonged below the thecae, which are ventrally unnapendaged, dorsally unnapendagged or with a minute tooth, which is bilobed or not, a glabrous ovary, and fruits with a persistent calyx. Other distinctive characters in the clade are the strongly decussate flattened young branches (in *M. michelangeliana* and *M. paradoxa*; lacking in *M. lucenae*), and 4-merous flowers with 4-celled gynoecia (in *M. lucenae* and *M. paradoxa*; 6-merous flowers with 6-celled gynoecia in *M. michelangeliana*). All three species occur in roughly similar latitudes, two

**Table 1** **Comparative features among *Miconia lucenae* and relatives in clade paradoxa plus *Miconia magnipetala*, a species that is morphologically similar, but not sampled in the phylogeny.** The table includes the Brazilian state to which they were recorded and vegetation type. The table does not include characters that are shared by all four species, such as the glabrous or glabrescent branches and leaves, white petals and yellow stamens, these with the connectives not prolonged bellow the thecae, glabrous ovary, and the fruits with a persistent calyx.

| Character/Species | *M. lucenae* | *M. magnipetala* | *M. michelangeliana* | *M. paradoxa* |
|---|---|---|---|---|
| Habit, plant size | small shrubs, up to 1.5 m tall | small shrubs, up to 1 m tall | trees , 8–12 m tall | small shrubs, up to 1.5 m tall |
| Young branches, shape | terete | terete | strongly flattened-decussate | strongly flattened-decussate |
| Number of bracetoles per flower | 2 | 4 | 2 | 2 |
| Pedicel | absent | 2–4,5 mm long | absent | absent |
| Flower, number of parts | 4-merous | 4-merous | 6-merous | 4-merous |
| Calyx tube/sepals inner portion margins | ciliate | glabrous | glabrous | glabrous |
| Calyx outer teeth | distinct | distinct | not perceptible | distinct |
| Petals shape/apex | lanceolate/broadly acute to narrowly rounded | broadly lanceolate/acute | oblong to oblanceolate/rounded | obovate/obtuse to rounded |
| Stamen connective, appendages | unnapendaged or with a small bilobed dorsal tooth | unappendaged | unappendaged or with two small dorsal teeth | unappendaged |
| Ovary, number of locules | 4-celled | 4-celled | 6-celled | 4-celled |
| Distribution/vegetation | Espírito Santo/rainforest | Espírito Santo/rainforest | Espírito Santo/rainforest | Minas Gerais/"campo rupestre" |

of them endemic to rain forests in Espírito Santo (*M. lucenae*, *M. michelangeliana*, the former also in Santa Teresa, to which *M. lucenae* is endemic), and one endemic to the neighboring state of Minas Gerais, but in "campos rupestres" (i.e., not in rain forests). The differences between the species in Paradoxa clade are summarized in Table 1. In addition to the other members of the Paradoxa clade, *Miconia magnipetala* (R.Goldenb. & Camargo) R. Goldenb. (formerly *Leandra magnipetala*; see *Camargo & Goldenberg, 2011*) is another species morphologically similar to *M. lucenae*. *Miconia magnipetala*, also endemic to forests of Espírito Santo, has not been sampled in our phylogeny, but it shares with *M. lucenae* 4-merous flowers, persistent sepals in the fruits, each with a distinct internal lamina and an acute external teeth, the broadly lanceolate petals, yellow stamens, and 4-locular ovaries. Despite the unknown phylogenetic position of *M. magnipetala*, and given some shared morphological traits and geographical distribution, it was also included in Table 1. The inclusion of *M. magnipetala* in the Paradoxa clade still needs to be tested in a phylogenetic context.

As for its placement in the traditional generic and infra-generic classification in Miconieae (i.e., *Cogniaux, 1891*), *M. lucenae* would fit in *Leandra* sect. *Oxymeris* (DC.) Cogn., due to the apical inflorescences, lanceolate petals and glabrous vegetative parts. No species in this genus and section has a combination of 4-merous flowers, yellow stamens and 4-celled ovaries. In terms of overall appearance, a species described in *Leandra* sect. *Oxymeris* that seems to be morphologically close to *M. lucenae* is *Leandra mourae* Cogn. (=*Miconia leamourae* R.Goldenb.), from montane forests in Rio de Janeiro. This species

was chosen by *Camargo & Goldenberg (2011)* for the diagnosis of *M. magnipetala* (see above), but it belongs to the Clidemia/Leandra clade (*Reginato & Michelangeli, 2016*). It also has vegetative features similar to *M. lucenae*, but it has 5-merous, pedicellate flowers with white stamens, longer hypanthia (2.5–3.5. mm long vs. 1.4–2 mm in *M. lucenae*) and external teeth on the sepals (1–2 mm long vs. ca. 0.2 mm).

## CONCLUSIONS

In large and diverse groups such as Miconieae, both morphological and geographical contexts are important to define species. Nevertheless, molecular data may allow to position the species among its relatives. In the case presented here, *Miconia lucenae* would be placed among species in *Leandra* s.s. clade due to morphological features and geographic distribution, but the phylogeny places it in a small clade, apart from the former.

## ACKNOWLEDGEMENTS

We thank Diana Carneiro for the drawings, and Paulo José Guimarães, Marie Claire Veranso-Libalah for their reviews and Mike Thiv for suggestions to previous versions of this manuscript.

### Funding
This research was funded by the NSF (DEB-1343612 & DEB-0818399) and received the support of Jayne and Leonard Abess of Fabián Michelangeli, who also received a visiting professor grant from CAPES and PRINT/UFPR (#88881.311854/2018-01). Renato Goldenberg receives a grant from CNPq/Brazil (Produtividade em pesquisa, #308065/2017-4). There was no additional external funding received for this study. The funders had no role in study design, data collection and analysis, decision to publish, or preparation of the manuscript.

### Grant Disclosures
The following grant information was disclosed by the authors:
NSF: DEB-1343612, DEB-0818399.
Jayne and Leonard Abess of Fabián Michelangeli.
CAPES.
PRINT/UFPR: #88881.311854/2018-01.
CNPq/Brazil: #308065/2017-4.

### Competing Interests
The authors declare there are no competing interests.

## Author Contributions

- Renato Goldenberg, Marcelo Reginato and Fabián A. Michelangeli conceived and designed the experiments, performed the experiments, analyzed the data, prepared figures and/or tables, authored or reviewed drafts of the paper, and approved the final draft.

## DNA Deposition

The following information was supplied regarding the deposition.

GenBank: Names after Miconia's new circumscription (Names before Miconia's new circumscription), in order of the 6 DNA regions: atpF-atpH; nrETS; nrITS; ndhF; psbK-psbI; trnS-trnG:

- Eriocnema fulva: MH743998.1; KF820735; MH743831.1; AY553781; KF821935; n.a.
- Miconia albicans: MH744009.1; KF820909; KF821554; EU055978; KF822092; n.a.
- Miconia antillana (Mecranium integrifolium): n.a.; KF820864; KF821531; n.a.; KF822051; n.a.
- Miconia ayisyena (Mecranium haitiense): n.a.; KJ933916; KJ933962; n.a.; KJ934016; n.a.
- Miconia bigibbosa (Conostegia bigibbosa): KM887065; KM893530; KM893587; n.a.; KM893674; KM893784
- Miconia blepharodes (Pleiochiton blepharodes): GQ139273; KF821342; GQ139302; GQ139316; GQ139330; KR062827
- Miconia calycina: n.a.; KF820956; EU055737; EU056001; KF822139; n.a.
- Miconia cinnamomifolia: MH744034.1; KF820982; EU055753; EU056013; KF822166; n.a.
- Miconia complanata: n.a.; KF821047; KF821615; EU055975; KF822231; n.a.
- Miconia crenata (Clidemia hirta): KR062211; KF820666; AY460479; n.a.; KF821866; KR062663
- Miconia crenulata (Clidemia crenulata): GQ139277; KF820637; EF418799; EU055910; GQ139333; n.a.
- Miconia dodecandra: MK296598.1; KF821020; KF821600; EU056026; JQ730527; MK296735.1
- Miconia friedmaniorum (Conostegia friedmaniorum): KM887086; KM893533; KM893628; n.a.; KM893722; KM893783
- Miconia gonoptera (Tococa gonoptera): n.a.; KF821384; AY460553; n.a.; KF822558; n.a.
- Miconia granatensis (Leandra granatensis): n.a.; KF820794; EU055691; EU055949; KF821985; n.a.
- Miconia grisebachiana (Calycogonium grisebachii): n.a.; KF820595; EU055646; n.a.; KF821796; EF549709
- Miconia hookeriana: n.a.; KF821059; EU055781; EU056040; KF822244; n.a.
- Miconia lima (Leandra lima): n.a.; KJ933953; KJ934006; EU055952; KJ934059; n.a.
- Miconia lucenae: MN557427; KF820763; KF821492; n.a.; KF821958; MN557428

- Miconia melastomoides (Leandra melastomoides): KR062278; KF820789; EF418830; n.a.; KF821980; KR062736

- Miconia michelangeliana: n.a.; KF821122; KF821655; n.a.; KF822309; n.a.

- Miconia myrtillifolia: KX073083; KX073119; KX073165; n.a.; KX073206; KX073187

- Miconia octona (Clidemia octona): n.a.; KF820678; KF821450; n.a.; KF821878; KR062664

Miconia paniculata: MF952906.1; MF953142.1; MF953156.1; n.a.; MF952921.1; MF952932.1

- Miconia paradoxa: n.a.; KF821148; n.a.; n.a.; KF822336; n.a.

- Miconia pepericarpa: MH744077.1; KF821153; KF821676; EU056071; KF822341; n.a.

- Miconia petropolitana: MH744079.1; KF821154; EU055815; EU056072; KF822342; n.a.

- Miconia reversa (Leandra reversa): n.a.; KF820828; EU055701; EU055958; KF822018; n.a.

- Miconia stenopetala (Leandra clidemioides): KR062237; KF820777; AY460540; EU055948; KF821968; KR062692

- Miconia subhisrsuta (Conostegia icosandra): KM887105; KF820719; AY460486; EU055933; KF821919; KM893781

- Miconia tococa (Tococa guianensis): n.a.; KF821385; AY460554; EU056136; KF822559; n.a.

- Miconia vulcanidomatia (Calycogonium rhamnoideum): n.a.; KF820605; KF821414; n.a.; KF821806; n.a

## Data Availability

Accession numbers for herbarium specimens (MBML: herbarium of the "Instituto Nacional da Mata Atlântica"; NY: herbarium of the New York Botanical Garden; RB: herbarium of the "Jardim Botânico do Rio de Janeiro"; UPCB: herbarium of "Universidade Federal do Paraná"; VIES: herbarium of "Universidade Federal do Espírito Santo"):

- Goldenberg 889, Paratype, MBML 33376
- Goldenberg 889, Paratype, NY 1015401
- Goldenberg 889, Paratype, UPCB 57441
- Goldenberg 1525, Holotype, UPCB 71800
- Goldenberg 1525, Isotype, MBML (deposited but not available online)
- Goldenberg 1525, Isotype, NY 2104713
- Goldenberg 1525, Isotype, NY 2104708
- Goldenberg 1525, Isotype, RB 14190053
- Kollmann 1973, Paratype, MBML 9016
- Kollmann 5484, Paratype, MBML 16165
- Kollmann 5484, Paratype, RB 520110
- Kollmann 5484, Paratype, UPCB (deposited but not available online)
- Kollmann 5494, Paratype, MBML 16266
- Kollmann 5494, Paratype, RB 519987

- Kollmann 5494, Paratype, UPCB (deposited but not available online)
- Kollmann 5494, Paratype, VIES 23010
- Kollmann 8160, Paratype, MBML 24837
- Kollmann 8160, Paratype, UPCB (deposited but not available online)

## New Species Registration

The following information was supplied regarding the registration of a newly described species:

Plant taxon: Miconia lucenae R.Goldenb. & Michelang 77206324-1.

## Supplemental Information

Supplemental information for this article can be found online at http://dx.doi.org/10.7717/peerj.8752#supplemental-information.

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
