# Peer review of "Miconia lucenae (Melastomataceae), a new species from montane Atlantic Forest in Espírito Santo, Brazil"

_PeerJ, doi:10.7717/peerj.8752_

## Round 0.1 · original submission · Minor Revisions

Dear Dr. Goldenberg,

Your ms. has received positive feedback. Rev. 2 has only minor points to change. In agreement with rev. 1 I also suggest to consider the latest taxonomy. Additionally, it would be indeed interesting to add the conservation status.

Best wishes
Mike Thiv

·

Basic reporting

The authors consistently present the description of a new species of Miconia, including comparison with closely related species and their position in phylogeny. The taxon is richly illustrated and is presented in real and potential geographical distribution.

Experimental design

nothing to add

Validity of the findings

The authors have long experience in the systematics of this plant group based on fieldwork and phylogenetic analysis.
In addition, the closely related species were satisfactorily compared, and the observed variations: morphological and geographical were tabulated.
Although these have recently published taxonomic changes, species are named by old names.
Despite their justification in Materials & Methods that "old names is more easily understandable", nomenclatural updates should be included in the paper. Included in the table with the Genbank access number.
I missed an analysis of the state of conservation of this new species. Although few individuals have been collected so far, there is no analysis of the degree of threat suffered. Even more so, it honors a researcher recognized for his work in conservation policy in the state of Espírito Santo.
The sequences of atpF-atpH and trnS-trnG of the new species are not available on genbank.
But they were sent as a complementary txt file.
>Miconia_lucenae_T813_atpF-atpH_MN557427
>Miconia_lucenae_T813_trnS-trnG_MN557428
Other comments and suggestions are presented in the text.

Additional comments

My biggest suggestions are for the inclusion of the new taxonomic combinations next to the old names, and the presentation of the state of conservation.

·

Basic reporting

The paper titled ''Miconia lucenae (Melastomataceae), a new species from montane Atlantic Forest in Espírito Santo, Brazil'', is well written. However there are typos e.g. see page 2 lines 28. I have made several comments in the document regarding this.
The literature cited is appropriate and update. However, the authors need to check the in text citation style in the guideline again. In several places the authors use et al. and & al. (e.g. compare lines 37-40 and 65) . All these have been highlighted in the text.
The figures, tables and supplementary data are well done. However the titles of the table 1 and figures 1 & 3 need to checked again.
Overall the paper is well written with just some minor things to check especially in text citation.

Experimental design

The methods and analyses provided are sound.

Validity of the findings

The authors have provided enough evidence to support the description of this new species Miconia lucenae.

---

## Round 0.2 · Minor Revisions

Dear authors

I went through your manuscript. Thank you for considering the changes suggested by the reviewers. I think they are adequately implemented.
I still recommend some minor changes in the text. I have corrected the word file which will be sent to the PeerJ staff.

Please check everything and resubmit this version. Then your manuscript should be acceptable.

Best wishes
Mike Thiv

---

## Round 0.3 · accepted · Accept

Dear Renato,
Thank you for your resubmission. I regard it as acceptable.
Please just implement the following minor changes:

l. 97 Clock and tree models were linked ...
l. 99 prior was set to the Birth and Death model, without calibration points.
l. 245 ventrally unappendaged, dorsally unappendagged or have with a minute tooth
l. 261 forests of
l. 285 placed among species in the Leandra s.s. clade due to morphological features and their geographic distribution, but the phylogeny places it in a small clade, apart from the former.

I hope that this new species will be adequately protected in this area...

Best wishes from Germany
Mike